# Study of the Influence of the Type of Aging on the Behavior of Delamination of Adhesive Joints in Carbon-Fiber-Reinforced Epoxy Composites

**DOI:** 10.3390/ma15103669

**Published:** 2022-05-20

**Authors:** Paula Vigón, Antonio Argüelles, Victoria Mollón, Miguel Lozano, Jorge Bonhomme, Jaime Viña

**Affiliations:** 1Department of Construction and Manufacturing Engineering, University of Oviedo, Edificio Departamental Oeste, 33203 Gijón, Spain; vigonpaula@uniovi.es (P.V.); antonio@uniovi.es (A.A.); lozanomiguel@uniovi.es (M.L.); bonhomme@uniovi.es (J.B.); 2Department of Materials Science and Metallurgical Engineering, University of Oviedo, Edificio Departamental Este, 33203 Gijón, Spain; mollonvictoria@uniovi.es

**Keywords:** composites, delamination, adhesive, fracture

## Abstract

This study analyzes the behavior under the static delamination and mode-I fracture stress of adhesive joints made on the same composite material with an epoxy matrix and unidirectional carbon fiber reinforcement and two types of adhesives, one epoxy and the other acrylic. Standard DCB tests (for mode-I fracture) were used to quantify the influence on the interlaminar fracture toughness of the type of adhesive used. Both materials were subjected to two different degradation processes, one hygrothermal and the other in a salt-fog chamber. After aging, the mode-I fracture has been evaluated for both materials. From the experimental results obtained, it can be deduced for the epoxy adhesive that exposure to the hygrothermal environment used moderately modifies its behavior against delamination, while its exposure to the saline environment produces a significant loss of its resistance to delamination. For the acrylic adhesive, the hygrothermal exposure causes an improvement in its delamination behavior for all the exposure periods considered, while the saline environment slightly modifies its behavior. There is, therefore, a clear influence of the type of aging on the fracture behavior of both adhesives.

## 1. Introduction

The study of the delamination in composite materials has been studied in depth in recent years. Inside this field and with the objective to increase the industrial implementation of these materials, the study of adhesive joints in composite reinforced with fiber has increased significantly due to the advantages it offers, as opposed to other more traditional methods of joining. Some advantages are the improvement of the strength-to-weight ratio, the acceptable load transfer through the adhesive, the facility to carry out maintenance and repair operations, the flexibility of the design and the lowest cost associated with industrial assembly processes.

When the adhesion is studied in relation to the delamination of composite materials, it is important to consider the adhesive and the mechanical properties of the materials to join, the contact surfaces that are part of the joint and the structural rigidity of the composite material itself. These types of joints, in general, cause increased plastic dissipation energy at the fracture of the adhesive joint [1,2]. The adequate treatment of the surfaces to join can improve the adhesion significantly, which can be achieved by modifying the physical and chemical properties of the joining surface [3,4,5,6], especially with fiber-reinforced polymers due to the low surface tension and the wettability they exhibit.

Other important parameters associated with the behavior of adhesive joints that are being studied on different materials, including composite materials, are the mechanical and physical characteristics of the elements that comprise the joint [7,8,9,10,11,12] and the influence in the type and rate of stress of an applied load [13,14,15,16,17].

Another line of work associated with the behavior of adhesive joints opposing initiation processes and the growth of delamination is the study of the joint properties, such as dimensional properties, and concerning the typology of the adhesives used, addressed with different test methodologies, using pure shear tests [18] or through fracture mechanics, under mode-II stress [19].

The behavior of adhesive joints on mixed materials [20,21] with the incorporation of nanofiber reinforcements [22] and under different types of fractures [23,24] is also being studied under different experimental methodologies. An important experimental effort is being made in the line of modifying the properties of the adhesive joint by incorporating additional elements that help to improve the adhesion or act as reinforcement of the joint against delamination. The incorporated particles range from carbon nanotubes (CNT) [25,26] to graphene nanoplatelets (GNP) [27].

One of the other lines of work in this field is analyzing, under different orientations, the behavior of the adhesive joints in composite materials subjected to different degradation processes, such as humidity and its effects in the process of delamination under pure fracture modes [28], exposure to a saline environment [29], the absorption of water in joints with hybrid composite materials [30,31], the effect of temperature [32,33] and the combination of the effects of humidity and temperature [34,35].

It is also important to point out the effort undertaken to provide the scientific and technical community the tools necessary for the calculation and testing of structures subjected to different load states and fabricated with technologies associated to composite materials joined by adhesives, as well as the necessary knowledge for the numerical simulation of its behavior. Some of the studies undertaken in the last few years develop interesting methodologies that will allow to address the calculation and/or the verification of structures joined by this methodology [36,37,38,39,40,41].

The objective of this study is to evaluate the influence that different environmental processes have on the behavior of adhesive joints against the phenomenon of delamination, on a composite material fabricated with an epoxy matrix and unidirectional carbon fiber reinforcement. In the execution of the adhesive joints, different commercial adhesives of the structural type have been used: one with an epoxy base of the same nature as the matrix from the composite material used as a substrate and another with an acrylic base. For the characterization of the joints’ resistance to delamination, the energy release rate reached by the joint under mode-I fracture stress has been taken as the study parameter. The influence of two different processes of degradation on an adhesive joint has been analyzed: one is hygrothermal and the other is the exposure to a saline environment and the influence that each type of adhesive has in the behavior against the joint delamination.

The study carried out is applicable to industries in which it is necessary to make composite joints, such as aeronautics and the automotive industry. It can also be useful in the case of structural component repairs.

The scheme followed in the work is as follows (Figure 1):

## 2. Materials

### 2.1. Composite Substrate

The composite material used as the substrate consisted of an epoxy matrix and a 0°-oriented unidirectional carbon fiber reinforcement, traded as MTC510-UD300-HS-33% RW. The laminate to be bonded was made by vacuum molding, following the manufacturer’s recommended thermal-curing cycle. Table 1 shows the mechanical properties of the composite obtained. Each substrate has 9 layers with a total thickness of 2.75 mm

A PTFE film of 12 μm of thickness is located between the substrates at one end of the layers and will act as the initiator of the delamination process.

The composite material laminates were mechanized via a diamond cut machine to obtain samples of 20 mm wide and 150 mm long. These, subsequently, were joined in pairs along the outer surface with the previously mentioned adhesives. A PTFE film of 12 μm of thickness has been placed between the substrates on an end that will act as the initiator of the delamination process. The placement of this film has generated an initial crack of 50 mm from the load line. The total thickness of each sample was 4.5 ± 0.1 mm. Figure 1 shows the schema of the specimen.

The thickness of the adhesives (Loctite and 3M) was measured on 3 different specimens of each of them. It was carried out with a ZEISS magnifying glass, model Stemi 508, which has an Axiocam 208 Color camera attached, and the measurements were taken with a magnification between 3.2× and 5×. The average values were 0.278 mm for epoxy adhesive and 0.215 mm for acrylic adhesive.

### 2.2. Characteristics of the Adhesives

Two commercial adhesives were used, one with an epoxy base and commercial reference Loctite^®^ EA 9461 and another with an acrylic base 3MTM DP8810NS, to join each of the parts that compose the final lamination and whose surface had been previously treated. In Table 2, the basic technical characteristics of the adhesives used are represented.

## 3. Experimental Methodology

### 3.1. Surface Preparation

The composite material used as substrate was surface conditions by means of manual sanding by abrasion with Al_2_O_3_ sandpaper and grain P220. To assess the variations of the surface roughness on the composite material before and after the sanding, roughness tester Mar-Surf M 300 = RD 18 was used, performing a sweep in two orientations: 0° (direction of the reinforcement fibers) and 90°. The considered parameters are as followed:

Ra: arithmetic mean of the absolute value of the roughness profile;

Rz: arithmetic mean of the consecutive distance between peaks and troughs;

Rmax: individual value of the maximum height between peak and trough.

Table 3 shows the surface condition before and after the surface treatment of the substrate. Once the surface of the composite was treated, it was cleaned and degreased with acetone, for the subsequent adhesive joining process [44,45].

### 3.2. Environmental Degradation Processes

The purpose of the degradation processes, called aging in this paper, used in this study was to assess the quality of the adhesive joint in function of time and of the effect of different external agents (humidity, temperature and saline concentration). For this, different simulated settings in chambers that accelerate the actions of these external agents were used.

These external agents that have been considered can affect the interface adhesive substrate as much as the individual components.

Knowing the effect that the external factors generate upon the joint allows to estimate its behavior in service, as well as helping in the appropriate selection of materials that comprise it and giving a solution to the possible problems that could make its industrial implementation costlier.

#### 3.2.1. Hygrothermal Aging

For the accelerated simulation of the hygrothermal process, a climatic chamber brand Vötsch, model VC 2020, was used. It is programmed to maintain constant environmental conditions during the time the samples are inside. In this case, the selected environmental conditions for this treatment were a constant temperature of 60 °C and a relative humidity of 70% [46,47]. The test sample’s times spent in the chamber were 1, 2, 4 and 24 weeks. In Figure 2, you can see the arrangement of some of the samples inside the climatic chamber.

#### 3.2.2. Aging Process in Salt-Fog Chamber

For the accelerated simulation of the aging process in a saline environment, a saline-fog chamber brand Köheler, model DCTC 1200 P, was used. The considered environmental conditions were an interior average temperature of 35 ± 2 °C, relative humidity of 89%, 1.2 bar air pressure and a saline solution prepared by dissolving sodium chloride “p.a” quality (reactive for analysis) in distilled demineralized water with a concentration of 50 g/L, a relative density between 1.0255 and 1.04 g/cm^3^, pH between 6.5 and 7.2 and a flow between 1 and 2 mL/h. At the end of the process, the samples were taken out, discarding the residues from the saline solution. The selected dwell times in the saline-fog chamber were 1, 2, 4, 12 and 24 weeks.

### 3.3. Characterization of the Behavior of Material against Delamination

To study the influence that the different aging processes have on the delamination process, the energy release rate under mode I of fracture were used as a study parameter. The samples of type DCB have been tested previously, and for the execution of the test, the methodology followed is the one proposed by the rule ASTM D5528-13 [48]. Piano hinges were used for the application of the load to the sample.

From the different proposed formulations by said rule, for the assessment of the energy release rate under mode I of fracture, G_IC_, the modified beam theory (MBT) has been employed. It uses the ensuing expression [48]:G_IC_ = 3Pδ/(2b(a + |∆|))
where b is the width of the test tube, P is the applied load, δ is the displacement on the application point of the load, a is the crack length and ∆ is a correction factor obtained as a function of the flexibility and length of the crack. The justification for this decision is based on the small difference of values obtained between the three basic formulations proposed by the rule. The test layout is shown in Figure 3.

All of our samples were tested using a servo hydraulic test machine (MTS 810) equipped with a load cell of 5 kN at a constant test velocity of 2 mm/min and at room temperature. A high-resolution camera monitored the advance of the crack. Two different positions of the crack tip can be seen in Figure 4.

Five specimens were tested for each material, each type of aging and each exposure period.

## 4. Results and Discussion

### 4.1. Hygrothermal Degradation Process

Figure 5 and Figure 6 present the evolution of the energy release rate under the mode-I fracture of the adhesives Loctite^®^ EA 9461TM, with an epoxy base, and 3M TM DP8810NS, with an acrylic base, respectively, over time in the hygrothermal chamber.

It is observed that the behavior of the adhesive joint against delamination is not altered by the humidity and temperature conditions during the first four weeks of exposure. For a longer period of exposure, 24 weeks, a loss of 7% is produced regarding the initial behavior which seems to indicate a small increase in the fragility of the adhesive as the time exposed to the thermal effect increases, considering that the saturation of humidity should be produced during the first days of exposure.

A different behavior is observed for this acrylic adhesive. In all exposure periods, the behavior of the adhesive joint improves with the dwell time in the chamber, obtaining in the first few weeks a higher energy release, 40%, compared to unaged reference joints. This tendency is maintained in longer periods of exposure, although with lower values. This behavior can be justified when the thermal process the joint has been submitted to has generated a postcure of the adhesive during the first two weeks and this way its behavior improves.

For longer exposures, a loss in tenacity would already be noticeable, but regarding the maximum values (two weeks), not regarding the initial value.

### 4.2. Degradation Process in Salt-Fog Chamber

Figure 7 and Figure 8 for the adhesives considered, Loctite^®^ EA 9461TM, epoxy, and 3M TM DP8810NS, acrylic, respectively, show the energy release rate under mode-I of the fracture in a function of the time spent in the saline-fog chamber for the two adhesives analyzed in this study.

For the epoxy adhesive, it can be seen that the conditions to which it has been subjected modify its behavior in regard to the original, with no exposure. The result is a loss of the delamination resistance capacity, except in the first week of exposure in which the energy release rates obtained are higher than the ones from a material with no exposure to this environment, 11%. From this moment on, a continued decline begins so that after 24 weeks, the loss is approximate to 20%.

Once again it is understood that the relatively increased values reached in the first week of exposure can be explained on the basis of the postcuring of said adhesive during this period of time.

The behavior of the acrylic adhesive is different, with a loss in its energy release rate that reaches up to 20% in the first two weeks of exposure to then subsequently relax that loss to 10% at 12 weeks of exposure. An improvement can also be seen at around the four-week mark, of up to 25%, possibly originated from the possible favorable effects that the temperature can generate on the adhesive, as would also take place in the first weeks of exposure on this adhesive subjected to a hygrothermal environment.

### 4.3. Comparison of Degradation Processes

Figure 9, for the epoxy adhesive, represents the energy release rate under mode I of the fracture as a function of different time periods of exposure, on one hand, to a saline environment and, on the other, a hygrothermal exposure.

In view of the obtained results of both environmental degradation processes to which the joints formed with the epoxy adhesive, fluctuations throughout the period of exposure can be observed of the energy release rate of the material under mode I of the fracture. For both processes, at the end of the selected dwell periods in the thermal and salt-fog chamber, it can be seen a loss of the resistance capacity of the joints generated with this adhesive. The reduction is less important in the case of the hygrothermal process and more important for the salt spray chamber process which seems to imply that the temperature does not significantly condition the behavior of the joining nor does the relatively low humidity generated in the hygrothermal process, with the saline environment being the one that can affect the behavior of the joint where the humidity is higher.

In Figure 10, for the acrylic adhesive, it shows the energy release rate under mode I of the fracture as a function of the different time periods of exposure considered for the types of chamber degradation they have been subjected to.

In this adhesive, acrylic, a different behavior to the epoxy adhesive is observed and, in this case, it is due to the environmental degradation process it was subjected to. The material subjected to a hygrothermal environment improves its behavior noticeably with all periods of exposure. The material subjected to exposure in the saline environment chamber, in general, suffers a loss of capacity against the delamination process in practically all periods of exposure, apart from the four-week exposure. All this seems to indicate that the exposure temperature dominates the behavior of the joint and the humidity affects the process primarily through the composite material that forms the substrate.

### 4.4. Influence of Laminate Flexibility

Figure 11 and Figure 12 represent the evolution of the flexibility C [δ/P] trend lines versus the length of the delamination in the crack growth phase for some of the test samples. These have been considered representatives of the materials’ behavior for the different exposure periods determined in the saline-fog chamber. Both analyzed adhesives have been represented.

In the case of exposure to a saline environment, it is observed that in both adhesives there is a lower flexibility of the tested samples as the permanence time in the chamber increases and that, in general, tends to stabilize as the exposure time increases. It is understood that this behavior stems directly from the behavior of the substrate and not of the used adhesive, meaning that it is in the base compound material in which different diffusion phenomena are produced that condition the behavior. In the case of loading the hygrothermal environment, there are no substantial differences in the inclines of the different trend lines, with practically a single line existing.

### 4.5. Fracture Surfaces

Once the samples had been tested, the resulting fractured surfaces were analyzed. Figure 13 presents a summary of the different superficial failures obtained, from some of the test samples considered to be more representative, according to the type of aging for each of the adhesives used.

There is a clear difference in the fracture surfaces between both of the adhesives analyzed, one epoxy and the other acrylic. It is observed that with the epoxy adhesives there is a tendency to a cohesive fracture with, in general, a presence of fiber bridges, whilst the acrylic adhesive presents mixed fracture surfaces, cohesive adhesives.

Thereby, for the epoxy adhesive, independently from the type of aging and period of exposure, in the delamination initiation zone, a cohesive-type failure occurs followed by a substrate failure, or interlaminar, followed in some zones with an adhesive failure, throughout the rest of its surface. This mechanism is primarily due to the formation of fiber bridges that displace the crack propagation line to the inside of the substrate, resulting in a crack advancement with fiber breakage, observed in many of the analyzed samples. In general, for both degradation processes applied, it is observed that the morphology of the fracture surfaces are not substantially modified throughout the different exposure periods. Although, in the case of aging by the salt-fog chamber, the decrease in the cohesive failure zone and the occurrence of adhesive failure points can be observed, although the characteristic interlaminar failure of the samples joined by this epoxy adhesive is still present.

For the test samples joined by acrylic adhesive, a fracture development with a mixed failure is observed, both adhesive and cohesive, over all of the test tube surface, which in this case is easily differentiable as it ends with the start of the cohesive failure, developed during the manual separation of substrates after the test’s completion. This mixed failure, predominant for this adhesive and for both degradation processes, can be understood when seeing that the images of the adhesive’s fractured surface does not display a complete symmetry between the parts of the test tube, the adhesive failure being located in the darkest areas and the cohesive in the lightest. Furthermore, it can be appreciated that the color changes on the adhesive edges, characteristic of a plastic deformation previous to the tear in a material with a more ductile behavior, have an effect that is more pronounced in the case of aging by a thermal chamber. A tendency to adhesive failure can also be appreciated in prolonged periods of aging in salt-fog chambers.

## 5. Conclusions

Two types of adhesive joints have been submitted to two different processes of degradation, a hygrothermal one and another being a saline environment, with the aim to analyze their behavior against delamination under load in mode I of a fracture. The most relevant conclusions achieved are:

For the epoxy adhesive, the exposure to the hygrothermal environment, to which the tested samples have been subjected, slightly modifies the initial characteristics of the adhesive joint, producing at the end of the exposure process a slight loss of resistance to delamination.

For the acrylic adhesive, the hygrothermal exposure results on the compound in an improvement of its behavior against delamination for all exposure periods considered.

The exposure of the test samples joined by the epoxy adhesive to the saline environment creates a relevant loss in its capacity of resistance to delamination, possibly originated by the phenomenon of the humidity diffusion in the interior of the substrate that makes up the joining.

For the acrylic adhesive, the exposure of the tested samples in a saline environment slightly modifies its behavior to delamination.

The epoxy adhesive, for both degradation processes, shows a predominantly weak behavior producing a sharp decline of the load after reaching the maximum value; it also presents a discontinued propagation of the crack, producing jumps that decrease the applied load, and fiber bridges are frequent. This behavior is maintained throughout the different periods and types of permanence in the chamber. This type of failure begins as a small adhesive failure that links with a cohesive one and ends in an interlaminar form. The motive of the decrease in the properties of this adhesive, especially in the case of stress in the salt-fog chamber, could be due to an infiltration of water molecules in the substrate that produce a loss of rigidity in the composite.

The acrylic adhesive presents a propagation of a continued and stable crack, with the presence of fiber bridges not being frequent in these elements.

Basically, in most specimens of this type, a first adhesive failure appears, together with a cohesive and a mixed zone in which the previous defects are combined, and this is maintained throughout the aging process.

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
