# Peer review of "Study of the Influence of the Type of Aging on the Behavior of Delamination of Adhesive Joints in Carbon-Fiber-Reinforced Epoxy Composites"

_materials, 2022, doi:10.3390/ma15103669_

Round 1

Reviewer 1 Report

This manuscript presents an experimental investigation into the effect of environmental aging (hygrothermal aging and salt spray aging) on the delamination behavior of CFRP laminates used as substrates of adhesive joints with a DCB configuration. The latter were fabricated using two different adhesives: epoxy and acrylic.  

General evaluation

As a general comment based on both the form and the logic of the testing procedure, the approach adopted for the analyses is quite rigorous, but the text requires to be strongly improved to support the experimental results. Indeed, I found the manuscript sometimes imprecise and confused, mainly owing to poor explanations and some vague sentences that need to be rephrased or clearly explained by the authors. I recommend a revision by a native speaker: the lack of fluidity of the text (e.g., too long sentences) contributes to making the manuscript occasionally chaotic and, thus, difficult to read.

I recommend the authors consider my comments and suggestions on aspects that, in my opinion, still prevent the original manuscript from being considered for publication in the Materials journal.

Specific comments

  • The authors should first explain the added value of their work: is there a specific application of the system they are studying?
  • Use verbs in the full form (e.g.: “it’s” should be “it is”)
  • The authors should decide “which English” to use (British or US?) and standardize it.
  • Use “joining” instead of “union” and “joint” instead of “bond”.
  • Use dots instead of commas to divide units from decimal digits.
  • Replace “acrylic based adhesive” and “epoxy based adhesive” with “acrylic adhesive” and “epoxy adhesive”
  • The description of the results is often only qualitative: please make it more quantitative and/or support it with literature evidence.
  • The manuscript lacks figures/schemes that can help the reader interpret the description: sometimes a picture is worth a thousand words.
  • It is recommended that the authors reference and include the following literature works:
  •    https://doi.org/10.1016/j.compositesb.2018.12.095: on the effects of
    surface treatments of CFRP bonded joints, with details on mechanical
    abrasion
    -    https://doi.org/10.3390/ma14061533: comparison between accelerated and
    natural aging of epoxy adhesive and CFRP material
    -    https://doi.org/10.1016/j.ijadhadh.2018.08.002: hygrothermal aging of
    polymer joints made using the 3M DP8810NS acrylic adhesive

Section 2 – Materials

  • The materials used are described below, both the type and the basic characteristics of the composite material used as a substrate and the types and characteristics of the adhesives.” Not needed.
  • "2.1. Type of composite material" --> “2.1. Composite substrate
  • The composite material used in this study, as substrate, is composed by an epoxy matrix and unidirectional carbon fibre reinforcements with the trade name MTC510-UD300-HS-33%RW, in Table 1 the mechanical properties of the laminate are shown. The laminate base of this study that subsequently will be bonded by means of the selected adhesives, has been made through vacuum molding using a thermal curing cycle recommended by the manufacturer of said material. The reinforcement fibres that comprise it were arranged in a unidirectional orientation at 0°C.”.  This might be simplified as follows: “The composite material used as the substrate consisted of an epoxy matrix and a 0°-oriented unidirectional carbon fibre reinforcement, traded as MTC510-UD300-HS-33%RW. The laminate to be bonded was made by vacuum molding, following the manufacturer's recommended thermal-curing cycle. Table 1 shows the mechanical properties of the composite obtained”.
  • The authors should indicate the number of fiber layers and the final thickness of the resulting laminate.
  • Table 1: What does “CV” mean?
  • Are you sure the PTFE film was 12 mm in thickness?
  • Table 2:
    • “150.000 a 250.000”: What does “a” mean?
    • Was shear strength measured experimentally? I suppose you reported the technical data provided by the manufacturer in the adhesive TDS. If you extracted shear strength from the TDS, you should indicate the substrate with which it was measured (e.g., the TDS of 3M 8810 provides several substrates including CFRP; please check)
  • The procedure followed to fabricate the joints and cure the adhesives should be described in Section 2.2. In particular, the authors should clearly indicate the thicknesses of the two adhesives and the method used to control this parameter.
  • Lines 108-113 should be moved to Section 2.1. Moreover, “A PTFE film of 12 m of thickness...” is clearly wrong!

Section 3 – Experimental methodology

  • "The most relevant aspects of the experimental program performed for the characterization of the adhesive bonds against the process of delamination under mode I of stress fracture, when subjected to different environmental exposure processes are described hereunder.” Not needed.
  • Why did the authors use P80-grit sandpaper? They should know that such grit is too rough for plastic materials, specially reinforced plastics! In fact, the risk is to expose and damage the fibers, emphasizing delamination of the laminate. A typical grit used for composites for pre-bonding abrasion with sandpaper is between 180 and 220. Alternatively, Scotch-Brite sponges are commonly used for fine interface abrasion.
  • Table 3, caption: Replace “Superficial” with “Roughness
  • Knowing the effect that the external factors generate upon the bond allows [one] to predict its behavior in service”. This is not entirely correct: accelerated aging does not allow predictions, but only estimates of actual in-service behavior based on limited laboratory experience. This aspect is critical to the interpretation of the data.
  • Lines 145 and 156: Change “ageing process” to “ageing
  • Lines 149 and 159: Change “ oC” to “°C
  • The number of replicates per sample must be indicated.
  • A figure/scheme showing the geometry of test specimens should be added. Why do the authors refer to the samples as “tubes”?
  • Line 173: Replace “GIC” with “GIC
  • The GIC equation should be referenced, and the values assumed for Δ indicated.
  • Line 177: “a is the length of the delamination of the crack” should be “a is the crack length
  • The advance of the crack was monitored by a high-resolution camera”. Some images illustrating crack nucleation and propagation should be shown in the Results section.

Section 4 – Results and discussion

  • The experimental results obtained in the study of the behavior of adhesive bonds subjected to different degradation processes will be presented hereafter.” Not needed.
  • Figures 3 and 4: Error bars must be checked
  • Is the 7% loss in GIC statistically different from the unaged reference?
  • Lines 208-209: Replace “[…] to the ones obtained in the bonds without being submitted to any hygro-thermal processes” with “compared to unaged reference joints
  • Line 210: Replace “lover” with “lower
  • The aging conditions (for both hygro-thermal and salt-spray ones) have already been defined, so you should avoid repeating them each time. Likewise, please simply refer to the adhesives as epoxy adhesive and acrylic adhesive.
  • Replace “in function to” with “as a function of

Author Response

Answer to reviewer 1

Specific comments

  • The authors should first explain the added value of their work: is there a specific application of the system they are studying?

A paragraph has been added in this line

  • Use verbs in the full form (e.g.: “it’s” should be “it is”)

Done

  • The authors should decide “which English” to use (British or US?) and standardize it.

Done

  • Use “joining” instead of “union” and “joint” instead of “bond”.

Done

  • Use dots instead of commas to divide units from decimal digits.

Done

  • Replace “acrylic based adhesive” and “epoxy based adhesive” with “acrylic adhesive” and “epoxy adhesive”

Done

  • The description of the results is often only qualitative: please make it more quantitative and/or support it with literature evidence.

Our view is that the discussion is quite quantitative as can be seen in the following paragraphs:

A different behavior is observed for this acrylic adhesive. In all exposure periods the behavior of the adhesive joint improves with the dwell time in the chamber, obtaining in the first few weeks a higher energy release, 40%, to the ones obtained in the joints without being submitted to any hygro-thermal processes

For the epoxy epoxy adhesive, Loctite® EA 9461TM, it can be seen that the conditions to which it has been subjected, a saline concentration of 50g/l, a constant temperature of 35°C and relative humidity of 89%, modify its behavior in regards to the original, with no exposure, resulting in a loss of delamination resistance capacity, except in the first week of exposure in which the energy release rates obtained are higher than the ones from a material with no exposure to this environment, in a 11%. From this moment on, a continued decline begins so that after 24 weeks, the loss is approximate to 20%.

Once again it is understood that the relatively increased values reached in the first week of exposure can be explained in basis of the post curing of said adhesive during this period of time.

The behavior of the acrylic adhesive is different, with a loss in its energy release rate that reaches up to 20% in the first two weeks of exposure, to then subsequently relax that loss until a 10% at 12 weeks of exposure, an improvement can also be seen at around the four week mark, of up to a 25%, possibly originated from the possible favorable effects that the temperature can generate on the adhesive, as would also take place in the first weeks of exposure on this adhesive subjected to a hygro-thermal environment.

  • The manuscript lacks figures/schemes that can help the reader interpret the description: sometimes a picture is worth a thousand words.

A schema has been included at the end of section 1.

  • It is recommended that the authors reference and include the following literature works:

   https://doi.org/10.1016/j.compositesb.2018.12.095: on the effects of
surface treatments of CFRP bonded joints, with details on mechanical
abrasion
-    https://doi.org/10.3390/ma14061533: comparison between accelerated and
natural aging of epoxy adhesive and CFRP material
-    https://doi.org/10.1016/j.ijadhadh.2018.08.002: hygrothermal aging of
polymer joints made using the 3M DP8810NS acrylic adhesive

Done

Section 2 – Materials

  • The materials used are described below, both the type and the basic characteristics of the composite material used as a substrate and the types and characteristics of the adhesives.” Not needed.

Done

  • "2.1. Type of composite material" -->1. Composite substrate

Done

  • The composite material used in this study, as substrate, is composed by an epoxy matrix and unidirectional carbon fibre reinforcements with the trade name MTC510-UD300-HS-33%RW, in Table 1 the mechanical properties of the laminate are shown. The laminate base of this study that subsequently will be bonded by means of the selected adhesives, has been made through vacuum molding using a thermal curing cycle recommended by the manufacturer of said material. The reinforcement fibres that comprise it were arranged in a unidirectional orientation at 0°C.”.  This might be simplified as follows: “The composite material used as the substrate consisted of an epoxy matrix and a 0°-oriented unidirectional carbon fibre reinforcement, traded as MTC510-UD300-HS-33%RW. The laminate to be bonded was made by vacuum molding, following the manufacturer's recommended thermal-curing cycleTable 1 shows the mechanical properties of the composite obtained”.

Done

  • The authors should indicate the number of fiber layers and the final thickness of the resulting laminate.

Done

  • Table 1: What does “CV” mean?

Done

  • Are you sure the PTFE film was 12 mm in thickness?

12 mm

  • Table 2:
    • “150.000 a 250.000”: What does “a” mean?

Table 2 has been modified

  • Was shear strength measured experimentally? I suppose you reported the technical data provided by the manufacturer in the adhesive TDS. If you extracted shear strength from the TDS, you should indicate the substrate with which it was measured (e.g., the TDS of 3M 8810 provides several substrates including CFRP; please check)

Done

  • The procedure followed to fabricate the joints and cure the adhesives should be described in Section 2.2. In particular, the authors should clearly indicate the thicknesses of the two adhesives and the method used to control this parameter.

Done

  • Lines 108-113 should be moved to Section 2.1. Moreover, “A PTFE film of 12 m of thickness...” is clearly wrong!

Done

  • Section 3 – Experimental methodology

 "The most relevant aspects of the experimental program performed for the characterization of the adhesive bonds against the process of delamination under mode I of stress fracture, when subjected to different environmental exposure processes are described hereunder.” Not needed.

Done

  • Why did the authors use P80-grit sandpaper? They should know that such grit is too rough for plastic materials, specially reinforced plastics! In fact, the risk is to expose and damage the fibers, emphasizing delamination of the laminate. A typical grit used for composites for pre-bonding abrasion with sandpaper is between 180 and 220. Alternatively, Scotch-Brite sponges are commonly used for fine interface abrasion.

P220 has been used

  • Table 3, caption: Replace “Superficial” with “Roughness

Done

  • Knowing the effect that the external factors generate upon the bond allows[one] to predict its behavior in service”. This is not entirely correct: accelerated aging does not allow predictions, but only estimates of actual in-service behavior based on limited laboratory experience. This aspect is critical to the interpretation of the data.

We have changed the word predict by estimate

  • Lines 145 and 156: Change “ageing process” to “ageing

Done

  • Lines 149 and 159: Change “ oC” to “°C

Done

  • The number of replicates per sample must be indicated.

It has been indicated in the text.

  • A figure/scheme showing the geometry of test specimens should be added. Why do the authors refer to the samples as “tubes”?

A figure with the dimensions of the specimens has been added and the word “tubes” has been changed.

  • Line 173: Replace “GIC” with “GIC

Done

  • The GICequation should be referenced, and the values assumed for Δ

The equation has been referenced although is the Standard mentioned previously.

As far as the D value is concerned, the procedure indicated in the standard is used to obtain it and each specimen has its associated D value according to its flexibility. Nothing is provided by putting the D values of the large number of specimens tested.

  • Line 177: “a is the length of the delamination of the crack” should be “a is the crack length

Done

  • The advance of the crack was monitored by a high-resolution camera”. Some images illustrating crack nucleation and propagation should be shown in the Results section.

A new figure has been included with two images of the crack tip in different positions.

Section 4 – Results and discussion

  • The experimental results obtained in the study of the behavior of adhesive bonds subjected to different degradation processes will be presented hereafter.” Not needed.

Done

  • Figures 3 and 4: Error bars must be checked

Both figures have been modified

  • Is the 7% loss in GICstatistically different from the unaged reference?

The loss is exactly 7.34% and the bars have been modified to avoid this problem.

  • Lines 208-209: Replace “[…] to the ones obtained in the bonds without being submitted to any hygro-thermal processes” with “compared to unaged reference joints

Done

  • Line 210: Replace “lover” with “lower

Done

  • The aging conditions (for both hygro-thermal and salt-spray ones) have already been defined, so you should avoid repeating them each time. Likewise, please simply refer to the adhesives as epoxy adhesive and acrylic adhesive.

All has been simplified

  • Replace “in function to” with “as a function of

Done

Reviewer 2 Report

Tittle:

The work could be interesting to readers if title can be revisited to be a full tittle not two sentences.

Abstract:

Abstract does not reflect the whole picture of work done chronologically. The second sentence-line13 is too long and it loses its meaning. this section is poorly written with 5 sentences.  line 21, sentence has no meaning or message. 

Introduction:

Lacks coherence and flow. References are not in order. This section needs rework. "Phenomenon" is overly used. Sentences too long.

Materials:

Not well presented, not clear on epoxy or acrylic materials which is presented.

Results:

There is mention of thermal stability, therefore Thermogravimetric analyses and FTIR could be considered for addition support on the results present.

However, this work needs extensive attention on presentation and language for consideration. This includes "conclusion that is disjoint. 

Reference:

References are not in order and confusing, e.g. 24 comes before 23 and other mistakes.

Author Response

Answer to reviewer 2

Tittle:

The work could be interesting to readers if title can be revisited to be a full tittle not two sentences.

Title has been reduced to a sentence

Abstract:

Abstract does not reflect the whole picture of work done chronologically. The second sentence-line13 is too long and it loses its meaning. this section is poorly written with 5 sentences.  line 21, sentence has no meaning or message. 

Abstract was rewritten

Introduction:

Lacks coherence and flow. References are not in order. This section needs rework. "Phenomenon" is overly used. Sentences too long.

Abstract has suffered several modifications. Also, the word “phenomenon” has disappeared and the order of references has been modified

Materials:

Not well presented, not clear on epoxy or acrylic materials which is presented.

The presentation of materials has changed

Results:

There is mention of thermal stability, therefore Thermogravimetric analyses and FTIR could be considered for addition support on the results present.

In this paper we have chosen the evolution of mechanical properties for analyzing the effect of ageing. The use of other types of analysis will be taken into account for posterior studies.

However, this work needs extensive attention on presentation and language for consideration. This includes "conclusion that is disjoint. 

Many changes have been carried out in the paper.

Reference:

References are not in order and confusing, e.g. 24 comes before 23 and other mistakes.

These mistakes have been corrected

Reviewer 3 Report

  1. In table 1, change the τmáx to τmax.
  2. Line 149, correct temperature to 60 °C. And correct the degree sign through the paper.
  3. What are the reasons to select the temperature of 60 °C and humidity of 70%? Authors need to give some reference for the selection of temperature and humidity for other cases too.
  4. It is recommended to add the time-moisture content absorption graphs for both cases and write down the water absorption behavior of composite materials.
  5. Line 184, change 5 KN to 5 kN.
  6. Line 184, how did the authors select the velocity rate of 2 mm/min?
  7. Figures 3 to 6: Authors need to discuss in detail the reasons for different responses.
  8. It is recommended to discuss in detail the water absorption process for all cases and for details refer to the following articles:
  • Experimental study of quasi-static and dynamic tensile behavior of epoxy resin under cyclic hygrothermal aging. Polymer Degradation and Stability, 2022. (https://doi.org/10.1016/j.polymdegradstab.2022.109940)
  • Enhanced Impact-Resistance of Aeronautical Quasi-Isotropic Composite Plates Through Diffused Water Molecules in Epoxy. Scientific Reports, 11, 1775, 2021.
  • Hygroscopic Effects on the Penetration-Resistance Behavior of a Specially-Orthotropic CFRP Composite Plates. Composite Structures, Vol. 176: 1073-1080, 2017

Author Response

Answer to reviewer 3

  1. In table 1, change the τmáx to τmax.

Done

  1. Line 149, correct temperature to 60 °C. And correct the degree sign through the paper.

Done

  1. What are the reasons to select the temperature of 60 °C and humidity of 70%? Authors need to give some reference for selection of temperature and humidity for other case too.

It has been referenced

  1. It if recommended to add the time-moisture content absorption graphs for both cases and write down the water absorption behavior of composite materials.

No such study has been carried out in this paper. The proposal will be taken into account in subsequent works.

  1. Line 184, change 5 KN to 5 kN.

Done

  1. Line 184, how did the authors select the velocity rate of 2 mm/min?

This is the test rate set by the standard.

  1. Figure 3 to 6: Authors need to discuss in detail the reasons for different responses.

This part has been modified

  1. It is recommended to discuss in detail water absorption process for all cases and for details refer to following articles:
  2. Enhanced Impact-Resistance of Aeronautical Quasi-Isotropic Composite Plates Through Diffused Water Molecules in Epoxy. Scientific Reports, 11, 1775, 2021.
  3. Hygroscopic Effects on the Penetration-Resistance Behavior of a Specially-Orthotropic CFRP Composite Plates. Composite Structures, Vol. 176: 1073-1080, 2017
  4. Hygro Effects on the Low-Velocity Impact Behavior of Unidirectional CFRP Composite Plates for Aircraft Applications. Composite Structures, Vol. 135: 276-285, 2016.

These articles have been included in the references.

Round 2

Reviewer 1 Report

The authors provided a revised manuscript in which all my suggestions and comments were met. I confirm my previous positive opinion about the scientific accuracy of the experimental scheme followed. Moreover, after the revision, the work appears clearer, and the experimental results are better discussed. In view of this, I consider the revised version of the manuscript appropriate for publication in the journal Materials.

Just only one remark:

Figure 1: the total thickness should be changed to 4.5 mm, as stated in the text

Reviewer 3 Report

The authors address all the comments so this paper is recommended for publication.